# Rethinking Resistant Hypertension

**DOI:** 10.3390/jcm11051455

**Published:** 2022-03-07

**Authors:** Gabrielle Bourque, Swapnil Hiremath

**Affiliations:** Department of Medicine, University of Ottawa, Ottawa, ON K1H 7W9, Canada; gabourque@toh.ca

**Keywords:** hypertension, resistant hypertension, apparent treatment-resistant hypertension, adherence, mineralocorticoid receptor antagonists, sodium-glucose cotransporter-2 inhibitors, review

## Abstract

Resistant hypertension is common and known to be a risk factor for cardiovascular events, including stroke, myocardial infarction, heart failure, and cardiovascular mortality, as well as adverse renal events, including chronic kidney disease and end-stage kidney disease. This review will discuss the definition of resistant hypertension as well as the most recent evidence regarding its diagnosis, evaluation, and management. The issue of medication non-adherence and its association with apparent treatment-resistant hypertension will be addressed. Non-pharmacological interventions for the treatment of resistant hypertension will be reviewed. Particular emphasis will be placed on pharmacological interventions, highlighting the role of mineralocorticoid receptor antagonists and sodium-glucose cotransporter-2 inhibitors and device therapy, including renal denervation, baroreceptor activation or modulation, and central arteriovenous fistula creation.

## 1. Introduction

Resistant hypertension (RH) is usually defined as blood pressure (BP) that remains above guideline-specified targets despite the use of three or more antihypertensive agents at optimal or maximally tolerated doses, with one of those agents preferably being a diuretic (see Table 1). It is not uncommon, being identified in 10 to 30% of hypertensive patients [1], and it is known to be a risk factor for cardiovascular (CV) events, including stroke, myocardial infarction (MI), heart failure (HF), and CV mortality, as well as adverse renal events, including chronic kidney disease (CKD) and end-stage kidney disease (ESKD) [2,3,4,5].

The first step in the management of RH is to exclude apparent treatment-RH, which may be caused by the white-coat effect, medication non-adherence, and therapeutic inertia. Furthermore, evaluation for secondary causes of hypertension, most commonly primary aldosteronism, should also be conducted. Only after careful exclusion of these factors can true RH be diagnosed. Based on a recent systematic review [6], the prevalence of apparent treatment-RH has been estimated at 14.7% of all hypertensive patients, while the prevalence of true RH has been estimated at only 10.3% of those same patients.

## 2. Diagnosis and Definitions

Both office and out-of-office BP measurements can be used for the diagnosis of hypertension. The cutoffs for diagnosis vary with the measurement method used. Furthermore, BP targets also vary depending on the patient’s comorbidities and estimated CV risk [5]. Therefore, the Hypertension Canada guidelines for the management of RH do not specify a BP cutoff for the diagnosis of RH, but rather emphasize the BP remaining above targets despite the use of three or more antihypertensive agents at optimal doses, with one of those agents preferably being a diuretic. On the other hand, other organizations have chosen different single cutoffs for the diagnosis (see Table 1 for details). The epidemiological data which links RH and adverse CV outcomes traditionally used a threshold of 140/90, and clinical trials in this area also use the same as an inclusion criteria. Hence the European guidelines stick to that threshold. However, since the American guidelines use a threshold of 130/80 as the treatment target (irrespective of comorbidities), the same has been used for the RH definition. Additionally, the American guidelines also consider ‘controlled RH’, which is when the BP is at target (≤130/80), albeit with four or more drugs.

Not only are accurate and standardized office BP measurements important in the diagnosis of RH, out-of-office measurements, usually with a 24-h ambulatory blood pressure monitoring (ABPM), are crucial to rule out white-coat hypertension (defined as elevated clinic BP readings but normal 24-h ABPM values), which, in one study [7], was present in 37.5% of patients diagnosed with RH on the basis of office BP measurements. Home BP monitoring is another option to ABPM for the exclusion of the white-coat effect.

Drug profiles should be reviewed for drugs known to cause increased BP, and those should be discontinued wherever possible prior to the diagnosis of RH. Additionally, clinical (or therapeutic) inertia, defined as the failure of health-care providers to initiate or intensify therapy when therapeutic goals are not reached [8], is fairly common, and should be excluded and addressed prior to diagnosing RH. The 3Ds of pharmacotherapy are important to consider: drug doses, duration of drug action, and the use of a diuretic. For diuretics, preferably thiazide-like (i.e., Chlorthalidone or Indapamide) should be used as they are more potent and longer acting, with clinical outcome data, compared to the widely used hydrochlorothiazide. 

## 3. Adherence 

Medication non-adherence is more common in apparent treatment-RH than is usually appreciated. In the published literature, its prevalence has been estimated to vary from 3 to 86% based on individual studies from a systematic review [9]. It can be assessed through indirect measures, such as questionnaires, self-reports, pill counts, rates of prescription refills, medication event monitoring systems (MEMS), and patient diaries, or direct measures, such as BP response to directly observed therapy (DOT) and therapeutic drug monitoring (TDM), which involves measuring the levels of antihypertensive agents in physiologic fluids such as blood and urine [10,11,12] (see Table 2). In the systematic review mentioned above, the prevalence of non-adherence varied from as low as 13 to 19% based on indirect measures such as self-reports, physician interviews, and rates of prescription refills, to as high as 45 to 49% based on direct measures such as DOT and blood and urine assays. In another recent systematic review, the prevalence of non-adherence in patients with apparent treatment-RH was 35% (95% confidence interval (CI) 25 to 46%), however, with significant heterogeneity identified among individual studies, this leads to an estimate of the prevalence of non-adherence ranging from 3 to 86%. This varied between 25% (95% CI 15 to 39%) using indirect methods of assessment and 44% (95% CI 32 to 57%) using direct methods of assessment. In this study, there was an association between higher prevalence of non-adherence with younger age, gender (higher prevalence of non-adherence in men compared with women), and higher rates of non-adherence were also seen in the more recent studies [13]. Efforts to diagnose and address non-adherence should be made in all patients with suspected RH. Arguably, given its high prevalence in this setting, assessment of adherence should take precedence over searching for secondary hypertension.

## 4. Secondary Hypertension and Drug-Induced Hypertension

Several drugs and substances can raise BP, and whenever possible, their use should be decreased or stopped. Common culprits include calcineurin inhibitors, licorice, erythropoietin, tyrosine kinase inhibitors, non-steroidal anti-inflammatory drugs, cocaine, amphetamines, oral contraceptive agents, other sympathomimetics, and corticosteroids. The commonest causes of secondary hypertension are primary aldosteronism, followed by renovascular hypertension. Other less common causes would include phaeochromocytoma, paragangliomas, and other endocrine causes of hypertension. Screening and workup for these conditions should follow the specific patient presentations.

## 5. Management of Resistant Hypertension 

### 5.1. General Principles

Patients with RH are heterogeneous from a pathophysiology perspective. A large component of these patients, even after excluding primary aldosteronism, have a low renin phenotype, and would respond to sodium restriction and escalation of natriuresis in some form. A smaller proportion may have a neurogenic component to hypertension, and may respond better to sympathetic blockade. Indeed, a condition labeled as ‘refractory hypertension’ has been described, which refers to patients whose BP is uncontrolled despite the use of four drugs, including a thiazide-like diuretic and spironolactone. In these patients, little evidence of volume expansion is seen, and the neurogenic component may be more important, and the use of sympatholytic agents may be more beneficial.

### 5.2. Nonpharmacological Aspects

Nonpharmacological interventions should be reviewed and reiterated as part of the management of RH. These include physical exercise, weight reduction, abstinence from alcohol or reduction in alcohol consumption, adherence to the Dietary Approaches to Stop Hypertension (DASH) diet, reduction in sodium intake, increase in dietary potassium intake (provided that the patient is not at risk for hyperkalemia), and stress management. These patients are often salt sensitive, and efforts to reduce salt intake even at this stage are useful. In a small crossover trial of 12 patients with RH, the baseline mean sodium intake was 195 mmol/day. On a 50 mmol/day diet, compared to 250 mmol/day, the office BP decreased by 22.7 mm Hg (systolic) and 9.1 mm Hg (diastolic).

Measures to improve medication adherence and the elimination of drugs and substances known to increase BP should be undertaken where possible. Management of medication non-adherence may include interventions such as reminders and pill packs as well as exploration into the reasons underlying non-adherence, which may include specific patient motives and beliefs pertaining to hypertension and its management [13].

### 5.3. Pharmacological Aspects

Pharmacological management of hypertension traditionally includes the use of angiotensin-converting enzyme inhibitors (ACEIs) or angiotensin receptor blockers (ARBs), calcium channel blockers (CCBs), and thiazide-like diuretics as first-line. This is commonly referred to as the A-C-D combination. The definition of RH requires three drugs at optimal doses including a diuretic, and would usually mean including antihypertensives from the A-C-D combination.

#### 5.3.1. Spironolactone

RH has been associated with inappropriately elevated aldosterone levels and activation of the sympathetic nervous system [1]. Several systematic reviews and clinical trial data have reported a reduction in BP in the RH population with the use of agents such as spironolactone, amiloride, alpha- (such as doxazosin) and beta- (such as bisoprolol) adrenergic antagonists, and clonidine, with spironolactone demonstrated to have the greatest BP-lowering effect in several studies [14,15,16,17,18,19]. In the Prevention and Treatment of Hypertension With Algorithm Based Therapy-2 (PATHWAY-2) trial, the effect of spironolactone on home BP was compared with those of bisoprolol, a beta-blocker, doxazosin, an alpha-1 blocker, and placebo in patients with confirmed RH. The reduction in home systolic BP compared with placebo at 12 weeks was greatest with spironolactone (−8.7 mmHg (95% CI −9.72 to −7.69 mmHg)). The reductions were −4.48 mmHg for bisoprolol (95% CI −5.50 to −3.46 mmHg) and −4.03 mmHg for doxazosin (95% CI −5.04 to −3.02 mmHg). BP control was achieved in 57.8% of patients taking spironolactone, 43.6% of patients taking bisoprolol, and 41.7% of patients taking doxazosin, compared with 24.4% of patients receiving the placebo drug [14]. However, this was a short trial (3 months follow-up), and there was no data on the effects of any of these agents on clinically relevant outcomes, such as CV events. This was also the case in the other studies mentioned above. The BP lowering effect is highest with spironolactone; nevertheless, possibly in keeping with the higher prevalence of primary aldosteronism in this population. In this line, subsequent analysis from the PATHWAY-2 study reported 25% of patients having an inappropriately elevated plasma aldosterone concentration, with a strong correlation between BP lowering effect and the aldosterone-renin concentration (r^2^ = 0.13, *p* < 0.0001). However, the superior benefits of BP lowering with spironolactone in PATHWAY-2 extended across the range of renin levels except amongst those with very high renin levels where beta blockers were more effective.

#### 5.3.2. Other BP Lowering Drugs

As is apparent, the RCTs discussed in the previous section are all of a short duration, up to three months, and hence there are scant data on the benefit of specific pharmacotherapies on clinically relevant outcomes (e.g., CV morbidity and mortality). An epidemiological study of about 9000 RH patients used propensity score matching to compare CV outcomes with aldosterone antagonists as reference compared to alpha and beta adrenergic antagonists. In this study, the latter two classes had a lower risk of adverse CV outcomes, with an HR of 0.68 for alpha and an HR of 0.81 for beta adrenergic antagonists, respectively. This result probably reflects residual confounding, but does not provide us confidence in the long term benefit of specific drug classes in RH. The Resistant Hypertension Optimal Treatment (ReHOT) trial included 187 patients with RHT randomized to spironolactone compared to clonidine as additional therapy. At three months, the mean change from baseline in office BP (in mm Hg) was similar between spironolactone (−15.1 for systolic BP and −7.7 for diastolic BP) and clonidine (−13.7 and −6.4 respectively). However the decrease in 24 h ABPM (in mm Hg) was greater with spironolactone (−11.8 for systolic BP and −6.3 for diastolic BP) than with clonidine (−7.3 and −3.9 respectively). Thus the overall literature supports that, following spironolactone, pharmacotherapy with the addition of alpha- and beta-adrenergic antagonists, or clonidine to the baseline regimen decreases BP significantly, with ongoing uncertainty about the superiority of specific drug classes for clinical outcomes.

#### 5.3.3. Newer Agents

Sodium-glucose cotransporter-2 (SGLT-2) inhibitors, or flozins, are another class of medications that are increasingly being recognized and utilized not only for their glucose lowering efficacy, but also due to their recently demonstrated beneficial effects on CV and renal outcomes [20,21,22,23]. These clinical outcome benefits now seem to be seen even in the absence of diabetes, specifically in patients with either proteinuria, CKD, or HF. In addition to glucose lowering, the BP lowering effects of flozins have been studied in multiple independent trials. This effect seems to be independent of the dose used (unlike the glycemic effect), and is seen regardless of the underlying BP-lowering concomitant medications being used.

The BP lowering effects of flozins, mainly empagliflozin, were especially pronounced in two specific populations of hypertensive patients, namely diabetics with lack of nocturnal dipping in BP and Black patients with type 2 diabetes. In a study by Kario et al., Empagliflozin reduced mean 24-h systolic BP by 7.7 mmHg (2.8 to 12.5 mmHg, *p* = 0.002) and mean 24-h diastolic BP by 2.9 mmHg (0.8 to 5.0 mmHg, *p* = 0.008) compared with placebo in patients with diabetes and uncontrolled nocturnal hypertension [24]. Ferdinand et al., meanwhile, reported a reduction of 8.39 mmHg (3.04 to 13.74 mmHg, *p* = 0.0025) in mean 24-h systolic BP and 7.43 mmHg (2.48 to 12.37 mmHg, *p* = 0.0036) and 4.25 mmHg (1.29 to 7.21 mmHg, *p* = 0.0053) in office systolic BP and diastolic BP, respectively, with Empagliflozin compared with placebo in African Americans with type 2 diabetes [25]. It is interesting to note that diabetics with a lack of nocturnal dipping in BP and Black patients have been shown to have increased sodium sensitivity and fluid retention as major contributors to hypertension. In that setting, flozins might be especially useful in lowering BP due to their effects on natriuresis and osmotic diuresis.

The BP lowering effects of flozins especially in RH were examined in a study by Ferreira et al., a *post-hoc* analysis of the EMPA-REG Outcome study [21,26]. In this study, RH was defined as uncontrolled BP (systolic BP ≥ 140 mmHg and/or diastolic BP ≥ 90 mmHg) despite the use of ≥three classes of antihypertensive medications, including a diuretic, or controlled BP but on ≥four classes of antihypertensive medications, including a diuretic. 22.5% of patients had presumed RH, which is similar to its prevalence in hypertensive patients [1]. 85.9% of those patients were on beta-blockers, 100% on diuretics, 96.6% on ACEIs/ARBs, and 17.2% on mineralocorticoid receptor antagonists (MRAs). Compared with placebo, empagliflozin reduced systolic BP by 4.5 mmHg (95% CI 3.1 to 5.9 mmHg, *p* < 0.001) and diastolic BP by 1.7 mmHg (95% CI 0.9 to 2.5 mmHg). The proportion of patients with presumed RH that achieved a systolic BP < 130 mmHg was also higher with empagliflozin compared with placebo (38% vs. 26% at week 12 of follow-up). In this trial, the use of empagliflozin in presumed RH was also associated with a significant reduction in CV death, CV death or hospitalization for HF, and incident or progressive nephropathy. A major limitation of this study is that it is a *post-hoc* analysis and therefore does not consider factors such as medication adherence, clinical inertia, accurate and standardized BP measurements, and white-coat hypertension in the definition of presumed RH, which, as discussed above, are essential components of the evaluation of RH [27]. Secondly, the effect of canagliflozin on systolic BP, including in patients with apparent treatment-RH (defined as BP ≥ 130/80 mmHg despite receiving ≥3 classes of BP-lowering drugs, including a diuretic), was assessed by Ye et al. in another *post-hoc* analysis of the CREDENCE trial [28]. In this study, the prevalence of apparent treatment-RH was 31.4% of patients at baseline. Canagliflozin reduced office systolic BP by 3.50 mmHg (95% CI −4.27 to −2.72 mmHg) at three weeks, and this effect was sustained over the duration of the trial. Furthermore, canagliflozin lowered systolic BP across all prespecified subgroups according to baseline BP, the baseline number of BP-lowering drugs, and the presence or absence of apparent treatment-RH. Canagliflozin was also found to have kidney and CV protective effects, as there was a reduction of the composite outcome of kidney failure (chronic dialysis, transplantation, or sustained estimated glomerular filtration rate (eGFR) < 15 mL/min/1.73 m^2^), a sustained doubling of serum creatinine, or death caused by kidney or CV disease of 30% in the canagliflozin group compared with the placebo group (HR 0.70 (95% CI 0.59 to 0.82). Similar to the effect on systolic BP, this effect was consistent across all prespecified subgroups [29,30]. Of note, the contribution of office BP reduction to the reduction in event rates was relatively small, suggesting that other mechanisms (potentially related to vascular protection and sympatho-inhibitory effects) may explain the beneficial effects of canagliflozin (and other flozins) on clinically significant outcomes.

These data support the emerging and potential role of flozins in RH, however one should note that trials of flozins with clinical outcomes explicitly in the RH population have not been done. The data of clinical benefit cited above, even in subgroups from CV outcome trials, are in patients with *apparent* RH, given the absence of ABPM as inclusion. Nevertheless, the clear outcome data support the use of flozins in RH when accompanied by standard indications for these drugs, such as DM, CKD, albuminuria, and/or HF.

## 6. Role of Device Therapy

The surgical interventions to reduce BP that have been investigated include lumbar sympathectomy, renal denervation, baroreceptor activation or modulation therapy, and central arteriovenous fistula creation. These interventions are summarized in Table 3.

Lumbar sympathectomy is of historical importance, since it was initially deployed when no effective treatment for hypertension existed. It effectively decreased BP (up to 70 mmHg in some cases), but was abandoned due to its association with serious side effects, including paralytic ileus, impotence, loss of sweating, loss of sensation, and death [31,32], and due to the advent of effective pharmacotherapy.

Renal denervation involves the ablation of sympathetic afferents in the renal artery, which in turn decreases efferent sympathetic tone, which is known to play a role in maintaining BP. Initial reports on the use of this technique showed effective reduction in BP, up to 20 to 30 mmHg, in patients with severe RH [33,34]. However, in the subsequent larger sham-controlled randomized controlled trial (RCT) Symplicity HTN-3, there was no significant difference between the two arms with regards to change in mean systolic BP at six months (difference of −2.39 mmHg but 95% CI −6.89 to 2.12 mmHg, *p* = 0.26). There was also no significant change in the 24-h ambulatory systolic BP between the two arms (−1.96 mmHg but 95% CI −4.79 to 1.06 mmHg, *p* = 0.98) [35]. Potential explanations for these discrepancies are thought to relate to differences in adherence to antihypertensive medications in the two arms as well as to varying levels of expertise relating to denervation in the Symplicity HTN-3 study. Other potential factors thought to have contributed to the discrepancy between the results of earlier studies and of the more recent Symplicity HTN-3 study pertain to the earlier studies and include regression to the mean in the uncontrolled studies (preferential selection of patients with very high BP, causing a fall in subsequently measured BP not explained by any therapy due to the natural variability in BP over time), differences in adherence to antihypertensives between the two arms in the unblinded controlled studies (including patients’ increasing adherence to their antihypertensive medications due to the belief that they have received denervation), the placebo effect (a patient’s blood pressure dropping from the belief that they have received denervation) in the unblinded controlled studies, and information bias leading to asymmetrical data handling in the unblinded controlled studies [35]. More recent sham-controlled trials of renal denervation used a multielectrode catheter or an ultrasound-based denervation procedure to ensure more adequate ablation in patients with mild hypertension either on no medications or three or less medications. Adherence was also ensured by using DOT prior to measurement of ambulatory BP. These trials have shown significant, albeit smaller reductions in ABPM with renal denervation. The magnitude of reduction in BP was smaller in these trials, between 5 and 7.4 mmHg for systolic BP, and between 4.1 and 4.4 mmHg for diastolic BP. Of note, these trials only had a follow-up period of two to six months, and they did not evaluate the change in clinical outcomes, including CV outcomes, with renal denervation compared with sham procedure [36,37,38]. It is also important to note that this intervention has a low risk of procedural complications in the RCTs. Of concern is a potential risk of shock with hypovolemia or hemorrhage in the absence of the counterregulatory effect of an activated sympathetic system, but this has not been reported as a common side effect of renal denervation in the clinical trials mentioned above.

Baroreceptor activation therapy (BAT) involves activation of the myogenic stretch reflex in the carotid body, which results in a reduction in central sympathetic activity and a lowering of BP. After an initial promising pilot RCT, the pivotal double blind Rheos RCT showed a mean decrease in office systolic BP of 16 +/− 29 mmHg compared with 9 +/− 29 mmHg in the control group with this technique at six months, but this was not statistically significant (*p* = 0.08). 54% of the patients in the BAT group were responders, compared with 46% in the control group, which was not statistically significant (*p* = 0.97). 25.5% of the study participants developed adverse events. The most frequent adverse event with this intervention was nerve injury at the time of device implantation with the first-generation device [39]. In this trial, both the long-term safety and efficacy of the first-generation Rheos system were demonstrated. However, short-term safety and efficacy could not be demonstrated due to trial design and BAT methodology. A novel endovascular baroreceptor amplification device, MobiusHD, which increases wall strain in the carotid sinus, is currently undergoing pilot trials. A proof-of-concept trial with this device was promising, as it showed a reduction in mean office BP by 24 and 12 mmHg for the systolic BP and diastolic BP, respectively, at 6 months, with statistical significance. Furthermore, mean 24-h ambulatory BP decreased by 21 and 12 mmHg (systolic BP and diastolic BP, respectively), within the same time period, with few adverse events [40]. A larger prospective, randomized, double blind, sham-controlled study using this device is currently under way (CALM-2 study, clinicaltrials.gov: NCT03179800), and aims to evaluate the safety and effectiveness of the MobiusHD system on ambulatory BP in patients with RH. The CALM-START study is another prospective, randomized, double blind, sham-controlled study investigating the efficacy and safety of the MobiusHD system on ambulatory BP in patients with RH (clinicaltrials.gov: NCT02804087). The Barostim Neo System is a second-generation minimally invasive unilateral single electrode device that has been developed and is currently being investigated, although mostly in the HF population. No RCT-level data is available for this device in the RH population. However, two RCTs are currently under way. The Nordic BAT study is a randomized, double blind, parallel-design clinical trial which will include 100 patients with RH and will evaluate the effect of baroreceptor activation therapy on 24-h systolic ambulatory BP at eight months of follow-up compared with continuous pharmacotherapy, with secondary endpoints including effects of BAT on home BP, office BP, and autonomic function (clinicaltrials.gov: NCT02572024). The Economic Evaluation of Baroreceptor STIMulation for the Treatment of Resistant HyperTensioN (ESTIM-rHTN) study is another randomized open-label trial of BAT which is currently under way (clinicaltrials.gov: NCT02364310). This trial will compare the cost-effectiveness of BAT with usual care.

Arteriovenous fistula creation with a central anastomotic device is another device therapy that has been studied for the treatment of RH. This technique involves adding a venous segment, therefore with low resistance and high compliance, to the central arterial system in order to reduce BP by exploiting natural mechanical effects. This technique was reported to reduce mean office systolic BP by 26.9 mmHg and mean systolic 24-h ambulatory BP by 13.5 mmHg in a small RCT, but procedural complications occurred in more than 50% of patients, including late ipsilateral venous stenosis requiring intervention in approximately a third of the study participants [41]. In addition, arteriovenous fistula is known to potentially lead to high-output HF in other settings, such as hemodialysis [42]. Therefore, this technique has been abandoned.

Therefore, while device therapy is promising for the management of RH based on available research data, its use is currently only supported in the setting of clinical research. Renal denervation, at the time of writing, is approved in many jurisdictions, but not in North America, though that is expected soon. Some controversy remains about the cost and the selection of the patient population for its use. Given the inconclusive available evidence for the use of other devices in the setting of RH at present, they cannot be recommended yet.

## 7. Future Directions and Newer Agents

From the discussion above, it is clear that we understand far more about the pathophysiology and therapeutic options in RH. At the same time, there are unmet needs, and we do need better data to guide therapy, as well as more options targeting pathophysiological options. Drugs targeting certain pathways (e.g., Angiotensin Receptor-Neprilysin Inhibitors, ARNi) and non-steroidal mineralocorticoid antagonists (e.g., finerenone) have been approved for specific clinical indications, but may have BP lowering and clinical outcomes in the RH population worth testing. Additional agents such as aldosterone synthase inhibitors and endothelin antagonists are undergoing clinical trials and may provide more arrows in our pharmacological quiver. Most importantly, there are several questions that still need to be answered with clinical trials. The ACD combination being superior to other combinations has not been tested in a clinical trial, and it is possible that the incorporation of spironolactone, especially in the low renin setting, may be superior even as a first-line drug. The *post hoc* analysis of flozins and their BP and clinical benefit should be confirmed with dedicated trials in the RH population. The use of pathophysiological and mechanistic information could even guide the choice of pharmacotherapy beyond the mere addition of additional agents. Lastly, we do need to develop mechanisms to address nonadherence, both in terms of improved diagnosis as well as effective management strategies.

## 8. Conclusions

RH is common and known to be a risk factor for CV and renal events. The first step in its management includes ruling out apparent treatment-RH, which may be related to the white-coat effect, medication non-adherence, and therapeutic inertia. Evaluation for causes of secondary hypertension should also be performed in the appropriate clinical setting. Patients with suspected or confirmed RH are best managed by a hypertension specialist. The management of confirmed RH includes nonpharmacological interventions, with emphasis on strategies to improve medication adherence as indicated, and pharmacological interventions. The backbone of pharmacological management of RH involves the use of the A-C-D combination, followed by MRAs such as spironolactone. Newer class of drugs, such as flozins, show some promise in post-hoc analyses of trials. While device therapy (renal denervation, BAT, and arteriovenous fistula creation) shows promise in clinical trials, no recommendations could be made for its use in this setting yet, but this remains an area of active investigation.

## Figures and Tables

**Table 1 jcm-11-01455-t001:** Comparison of existing guidelines for diagnosis of resistant hypertension.

Guideline	ESH/ESC 2018	AHA-ACC 2018	Hypertension Canada 2020
BP Threshold	SBP > 140 and/or DBP > 90	SBP > 130 and/or DBP > 80	Above target
Number of anti-hypertensive medications	≥three optimally tolerated or best tolerated	≥three maximum or maximally tolerated, appropriate dosing intervals	≥three drugs from different classes, at optimally tolerated dosages, used simultaneously
Class of anti-hypertensive medications	ACEi/ARB, CCB, diuretic	three different classes, commonly ACEi/ARB, CCB, diuretic	three or more drugs of different classes, preferably including a diuretic
Method of BP measurement	Confirmed with ABPM or HBPM	Consider ABPM or HBPM	Confirm with ABPM
Adherence	Confirmed	Assess	Assess

BP: blood pressure; SBP: systolic blood pressure; DBP: diastolic blood pressure; ESH/ESC: European Society of Hypertension/European Society of Cardiology; AHA-ACC: American Heart Association—American College of Cardiology; ACEi: angiotensin converting enzyme inhibitor; ARB: angiotensin receptor blocker; CCB: calcium channel blocker; ABPM: ambulatory blood pressure monitoring; HBPM: home blood pressure monitoring.

**Table 2 jcm-11-01455-t002:** Methods to assess medication adherence.

Method	Strengths	Limitations
Indirect Methods		
Physician perception	Simple	Poor capacity of perception
Self-report/Report by a proxy/Patient diaries/Physician interviews	Inexpensive Suitable for routine clinical use	Tend to overestimate adherence Potential for data manipulation
Questionnaires (i.e., Adherence to Refills and Medications Scale, MMAS-4, MMAS-8, MARS)	Inexpensive Suitable for routine clinical use Qualitative data Educational value	Tend to overestimate adherence Potential for data manipulation
Pill counts	Inexpensive Suitable for routine clinical use Give information on non-adherence for particular medications	Time-consuming Potential for data manipulation Do not capture timing of medication intake
Electronic drug monitors/Electronic pillboxes/MEMS	Capture timing of medication intake May serve as adherence reminders	Expensive Potential for data manipulation Risk of electronic malfunction Unsuitable for routine clinical use Lack of multi-medication monitoring systems
Digital sensors	Capture timing of medication intake	Expensive Risk of electronic malfunction Unsuitable for routine clinical use Intrusive Not approved for hypertension treatment
Rates of prescription refills/Proportion of days covered	Inexpensive Suitable for routine clinical use Provide data on prevalence of adherence during a given period	Tend to overestimate adherence Do not capture timing of medication intake Require integration of all electronic pharmacy records Methodological issues in calculating non-adherence
Direct Methods		
DOT	Less potential for data manipulation Associated with improved BP control in the short term Ensures administration of correct medication in correct dosage at correct hours Suitable for confirming the effect of pharmacologic treatment (in combination with subsequent ambulatory BP measurement)	Expensive Time-consuming Resource-consuming Unsuitable for routine clinical use Subject to “white-coat adherence” Intrusive Cannot easily detect partial non-adherence Cannot identify particular drugs Risk of adverse reactions (including severe hypotension) in partially or completely non-adherent patients
TDM	Less potential for data manipulation Highly sensitive Potential BP lowering effect Provides precise data on individual drug adherence Can assess whether prescribed drug dosing generates adequate serum concentration Covered by most health insurance plans	Expensive Time-consuming Resource-consuming Does not capture timing of medication intake Unsuitable for routine clinical use Subject to “white-coat adherence” Only provides data on most recent ingestion Subject to differences in drug absorption and metabolism Cannot easily differentiate between adherence, partial non-adherence, and complete non-adherence Does not inform about clinical impact of non-adherence

Abbreviations: BP, blood pressure; DOT, directly observed therapy; MARS, Medication Adherence Report Scale; MEMS, Medication Event Monitoring Systems; MMAS, Morisky Medication Adherence Scale; TDM, therapeutic drug monitoring.

**Table 3 jcm-11-01455-t003:** Summary of device therapies: effect on BP, possible adverse effects, and current status.

Intervention	Effect on BP	Possible Adverse Effects	Current Status
Lumbar sympathectomy	Up to 70 mmHg decrease in some cases	Paralytic ileus, impotence, loss of sweating, loss of sensation, death	Abandoned
Renal denervation	Up to 20 to 30 mmHg decrease in initial reports Sham-controlled RCT Symplicity HTN-3: no significant difference between arms in mean systolic BP and 24-h ambulatory systolic BP More recent sham-controlled RCTs: 5 to 7.4 mmHg/4.1 to 4.4 mmHg decrease in 24-h ambulatory systolic/diastolic BP	Low risk of procedural complications Potential risk of shock with hypovolemia or hemorrhage	Approved in Europe and some other countries Not approved in North America as of 2021
Baroreceptor activation or modulation	Pivotal RCT: up to 16 mmHg decrease in office systolic BP compared with 9 mmHg decrease in control group Rheos RCT: no significant difference between arms in office BP MobiusHD system proof-of-concept trial: 24 mmHg/12 mmHg decrease in mean office systolic/diastolic BP and 21 mmHg/12 mmHg decrease in mean 24-h ambulatory systolic/diastolic BP	Nerve injury or damage (including to facial nerve) at time of implantation, dysphagia, paresthesias	MobiusHD system: two sham-controlled studies in patients with RH currently under way Barostim Neo system: currently being investigated, but mostly in the HF population. Two RCTs in patients with RH currently under way
Central arteriovenous fistula	26.9 mmHg decrease in mean office systolic BP and 13.5 mmHg decrease in mean 24-h ambulatory systolic BP	Procedural complications, including late ipsilateral venous stenosis requiring intervention High-output HF	Abandoned

Abbreviations: BP, blood pressure; HF, heart failure; RCT, randomized controlled trial; RH, resistant hypertension.

## Data Availability

Not applicable.

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
