# Peer review of "Rethinking Resistant Hypertension"

_jcm, 2022, doi:10.3390/jcm11051455_

Round 1

Reviewer 1 Report

The topic is very interesting and actual. The resistant hypertension is a constant challange. 

The article is well written. But, because the title of the article is “State of the art review”, I have some suggestions for completing it:

  • Resistant hypertension and associated phenotypes - what are controlled resistant hypertension, refractory hypertension;
  • Diagnostic approach and the importance of ABPM / HBPM in detection of RH;
  • Most frequent causes of secondary hypertension and drug-related resistant hypertension (short);
  • Therapeutic approach:
    • The substrate for pharmacological interventions (SNS, RAAS, Na retention…);
    • Other pharmacological options beyond the A-C-D algorithm – more details about β-blockers, α-β blockers, α2 agonists, α1 blockers, peripheral vasodilators;
    • New molecules / the future?
      • ARNi/finerenone for kidney protection/other?

The chapter dedicated to SGLT2i is well systematized, but it does not find its purpose in a review dedicated to RH. To date, there are no studies dedicated to this pathology, only post-hoc analyzes or studies that have analyzed the antihypertensive effect of flozins in general. I would suggest a shortening of this chapter.

Other suggested minor correction are putting directly in the article (attached).

Author Response

We would like to tell the reviewer for that thorough read and very helpful comments.
1.  We have added a discussion about the different definitions, including control resistant hypertension and refractory hypertension

  1. In the diagnostic approach, we have added the importance of home and ambulatory blood pressure monitoring

  2. We have added a paragraph on secondary hypertension, including drug related hypertension

  3. We have added a section on the biological rationale for drug choice in resistant hypertension

  4. We have added a paragraph on newer molecules in the future, including ARNi and Finerenone
  5. Lastly, we have shortened the section on the SGLT2 inhibitors.

7.  We also reviewed the direct comments made, and have corrected the typos and citations as pointed out by the reviewer.

Reviewer 2 Report

Review

In this paper entitled « Resistant hypertension: a state of the Art Review » the authors write paper about a very important topic.

Resistant hypertension is not uncommon as it may represent around 10% of all the hypertensive patients, so millions of patients.

This paper is well written and very pleasant to read with good references. However, I feel a little bit confused and disappointed by some points:

  • The paragraph about Spironolactone on page 3:
    • « RH has been associated with inappropriately elevated aldosterone »: this is true in around 20% of cases. But in 40% cases even though aldosterone is normal the renin is low (low renin HTN) indicating that sodium retention is the problem. This knowledge is the basis of the choice of spironolactone as the 4th drug.
    • I know there are still fear about the use of spironolactone, especially in patients with CKD or in male patients due to secondary effects such as gynecomastia but in my opinion the superiority of spironolactone in the lowering BP effect in Pathway-2 and the already known good effect of spironolactone in other conditions make this drug the 4th line drugs in RH (after CCB, RAS blocker and diuretic) and the beta blockers or alpha blockers should be used only in case of contraindication for spironolactone (or eplerenone or amiloride).
    • In the PHARES study, it has been shown that a drug treatment with the use of multiple diuretics acting in different tubular sodium reabsorption sites is better for the control of HTN than adding beta blockers (Bobrie et al. Journal of Hypertension 2012 Aug;30(8):1656-64)
    • Spironolactone may expose to increase of creatinine and K+ (it’s around 0.3 mmol/L for K+ and 10 µmol/L for creatinine, in the studies, in selected patients, mostly with eGFR > 45 AND K+<4.5 before spironolactone) and can lead to gynecomastia in around 7% of patients with doses < 50 mg (Jeunemaitre et al. Am J Cardiol 1987;60:820), but beta blockers and alpha blockers can also lead to side effect and in Pathway 2 the tolerance is better with spironolactone.

Creatinine baseline (µmol/L)

Creatinine final (µmol/L)

Mean difference

p

ASCOT

98,0

111,2

13,2

<0,001

ASPIRANT

81,0

88,0

7,0

<0,001

Oxlund et al.

81,2

87,2

6,0

<0,001

PATHWAY-2

84,79

93,21

8,43

<0,001

K+ baseline (mmol/L)

K+ final (mmol/L)

Mean difference

p

ASCOT

4,13

4,54

0,41

<0,001

ASPIRANT

4,15

4,52

0,37

<0,001

Oxlund et al.

3,9

4,16

0,26

<0,001

PATHWAY-2

4,06

4,49

0,43

<0,001

PATHWAY 2

Placebo

Spiro

Doxa

Bisoprolol

p

Serious AE

2%

2%

2%

3%

0,82

All AE

15%

19%

23%

23%

0,036

Stop for AE

1%

1%

3%

1%

0,28

      • BB may be associated with inferior protection against stroke risk and all-cause mortality when compared with other antihypertensive drugs (Thomopoulos et al. J Hyperten 2020;38(9):1669).
      • Alpha blockers may increase risk of heart failure and may also lead to higher rate of CV events (Davis et al. Ann Intern Med 2002;137:313 and ALLHAT JAMA 2000;283(15):1967).
      • The fact that the BP lowering effect is higher with spironolactone because there is a high prevalent of primary aldosteronism (PA) in RH is not a problem. On the contrary it is an argument for the use of spironolactone because we know that PA is under-screened and the main goal is not to be screened for PA but to have BP controlled (that is generally achievable with spironolactone in the vast majority of PA). The low renin state and PA is a continuum and in the 40% with low renin state the addition of spironolactone make sense and is efficient. Furthermore, in the Pathway 2 study the authors show that BP reduction is better with spironolactone than with the other drugs in 97% of cases!
      • The ESH 2018 Guidelines says “Thus, bisoprolol and doxazosin have an evidence base for the treatment of resistant hypertension when spironolactone is contraindicated or not tolerated” and the AHA 2018 guidelines says “Considerable evidence indicates that the addition of spironolactone to multidrug regimens provides substantial BP reduction when compared with placebo. Substantial data also demonstrate the advantage of spironolactone as compared with other active drugs. In particular, the recent PATHWAY-2 (Optimum Treatment for Drug-Resistant Hypertension) RCT demonstrated the superiority of spironolactone over alpha and beta blockers. »
      • So I think the conclusion of the authors about spironolactone is a little bit confusing.
  • The paragraph about « newer agents » (Page 3-4-5-6-7)
    • The SGLT2 inhibitors are definitely good drugs that we use more and more in many conditions and that have demonstrated beneficial effects on CV outcomes. However, 5 pages about this medication in paper entitled « Resistant hypertension: A state of the art review » is too long and I think totally out of the topic.
    • The first 2 paragraphs of the topic are about the effect of SGLT2 inhibitors with no link with HTN.
    • Then the third and fourth paragraph and the table 3 is very interesting but talking about HTN and not RHTN.
    • The 4th and 5th paragraph about the study by Ferreira et al (REF 38) and Ye et al (REF 39) are interesting and are about the topic of RH.
    • As SGLT 2 inhibitors are diuretics the conclusion of this paragraph could be that the use of diuretic is efficient in RH and it is in accordance with what has been shown with spironolactone. A nice RCT would be to compare the effect of spironolactone versus SGLT 2 inhibitors. Another nice RCT could be to see the effect of addition of SGLT2 as the 5th drug after CCB/Thiazide (chlortalidone or Indapamide)/RAS blocker/Spironolactone.
    • The PHARES study shown the good effect of adding different diuretics. The side effect and the security problem can be criticized (in case of diarrhea or heat wave for example) (Bobrie et al. Journal of Hypertension 2012 Aug;30(8):1656-64)
    • I think this paragraph about SGLT2 inhibitors should be shorten and I would advise the author to write another paper about the topic « SGLT2 inhibitors and Hypertension » with all the very good job already down (especially with the table 3) and I would advise the editor to consider this paper for publication in the journal as I think it would be very interesting.
  • The paragraph about “The role of device therapy” (Pages 7-8-9)
    • To the editor: it would be better to have the table 4 on the same page and no cut and the columns are to small it is a little bit difficult o read
    • I am not sure it is useful to mentioned in the table the lumbar sympathectomy and the central arteriovenous fistula (what is written in the text is enough)
    • Maybe the table could have more information, such as the table 3 with all the results of the study of renal denervation in RHTN
    • It is a little bit confusing because in the text the authors write 10 lines about SYMPLICITY HTN 3 which has negative results but less is say about the others study who show renal denervation can be efficient: DENER HTN, RADIANCE TRIO, SPYRAL … The authors talk about the RADIANCE SOLO: it is a very interesting study because it shows that RD works as I t can lead to BP control without drugs in 20% of patients but it must be emphasized that this study is not about RHTN. RADIANCE TRIO is about RHTN
  • Other comments:
    • The American definition of RH include patient who are controlled with four or more drugs maybe it should be written and commented
    • In the definition of RH the authors use the word “preferably” about the use of the diuretic in the 3 treatment (A-C-D). In the ESH 2018 guidelines and AHA 2017 it is written “should include a diuretic” and I think it is more appropriate (in my opinion we should write “must include a diuretic” …)
    • In the introduction: “evaluation for secondary causes of hypertension, including obstructive sleep apnea” : it is still a matter of debate to know if OSA is a cause of secondary HTN
    • About secondary HTN: I think the author should write a paragraph about “Secondary RHTN” and then explained that the paragraph 4 (Management and 5 (Pharmacological aspects) are about “Primary (=essential) Resistant Hypertension”
    • I think there is an error in the edition page 3: the “5. Pharmacological aspects” should be a sub title of the paragraph “4. Management of RH” and therefore should be written in italic, without number like “Nonparmacological aspects”
    • I agree with the ESH 2018 and AHA 2018 Guidelines: ABPM of HBPM can be both be used to rule out white coat hypertension so I wouldn’t say that home BP is an alternative, I consider they are equal.
    • In the table 1 the BP threshold is > or = (and not >)
    • The paragraph about adherence is interesting. In the paragraph it is written “Table 1” but it’s “Table 2”.
    • The Table 2 is interesting but why is it at the end of the article, after the references? If the article is shortening with less words written about SGLT2 inhibitors it could be intersitng to write more about adherence in RH.
    • In the conclusion: I appreciate that Spironolactone is cited and not alpha or Beta blockers but I think that it is premature to put spironolactone and flozins in the same level of evidence.

Author Response

We would like to thank the reviewer for the detailed and thorough review of our manuscript.
1.  We do agree with the reviewer about the choice of spironolactone as the fourth drug.  We also agree about the pathophysiology behind this choice and the evidence about its utility.  We have change the language to clarify this aspect better and expanded on the need for using spironolactone.  We have also included the suggested citations as the reviewer discussed.

  1. In agreement with the first reviewer as well, we have shortened the section on SGLT2 inhibitors to make it more coherent.
  2. On device therapy, we have included discussion about Radiance trial instead of Radiance solo.
  3. We have clarified the definitions, including the discussion about controlled resistant hypertension, and using the precise language from the guidelines
  4. We have added a paragraph on secondary hypertension prior to the discussion about management and pharmacotherapy

Round 2

Reviewer 2 Report

The authors answered my comment and they made changes in the manuscript, which is now better.